# UAV-YOLOv8: A Small-Object-Detection Model Based on Improved YOLOv8 for UAV Aerial Photography Scenarios

**DOI:** 10.3390/s23167190

**Published:** 2023-08-15

**Authors:** Gang Wang, Yanfei Chen, Pei An, Hanyu Hong, Jinghu Hu, Tiange Huang

**Affiliations:** Hubei Key Laboratory of Optical Information and Pattern Recognition, School of Electrical and Information Engineering, Wuhan Institute of Technology, Wuhan 430205, China; wanggang@stu.wit.edu.cn (G.W.); anpei@wit.edu.cn (P.A.); hhyhong@wit.edu.cn (H.H.); jinhuhu@stu.wit.edu.cn (J.H.); huangtg@wit.edu.cn (T.H.)

**Keywords:** UAVs, small-object detection, YOLOv8, WIoU, BiFormer, FasterNet

## Abstract

Unmanned aerial vehicle (UAV) object detection plays a crucial role in civil, commercial, and military domains. However, the high proportion of small objects in UAV images and the limited platform resources lead to the low accuracy of most of the existing detection models embedded in UAVs, and it is difficult to strike a good balance between detection performance and resource consumption. To alleviate the above problems, we optimize YOLOv8 and propose an object detection model based on UAV aerial photography scenarios, called UAV-YOLOv8. Firstly, Wise-IoU (WIoU) v3 is used as a bounding box regression loss, and a wise gradient allocation strategy makes the model focus more on common-quality samples, thus improving the localization ability of the model. Secondly, an attention mechanism called BiFormer is introduced to optimize the backbone network, which improves the model’s attention to critical information. Finally, we design a feature processing module named Focal FasterNet block (FFNB) and propose two new detection scales based on this module, which makes the shallow features and deep features fully integrated. The proposed multiscale feature fusion network substantially increased the detection performance of the model and reduces the missed detection rate of small objects. The experimental results show that our model has fewer parameters compared to the baseline model and has a mean detection accuracy higher than the baseline model by 7.7%. Compared with other mainstream models, the overall performance of our model is much better. The proposed method effectively improves the ability to detect small objects. There is room to optimize the detection effectiveness of our model for small and feature-less objects (such as bicycle-type vehicles), as we will address in subsequent research.

## 1. Introduction

With the continuous reduction in the production cost of UAVs and the gradual maturity of flight control techniques, the application of UAVs is becoming increasingly widespread in areas such as power line inspections [1], traffic monitoring [2], and crop analysis [3]. Object detection plays an increasingly important role as a key link in the missions carried out by UAVs and has very great research significance. The UAV has a wide field of view due to its high flying altitude, leading to problems such as a high proportion of small objects and complex backgrounds in the captured images, which increases the difficulty of the object detection task. Moreover, UAV platforms have limited resources, making it hard to embed high computational and storage-demanding object detection models. Therefore, enhancing the performance of object detection while considering the limited resources of the hardware platform is one of the core issues in object detection in UAV aerial scenes.

The essential difference among object detection algorithms is that the features of the image are extracted differently. Most of the traditional object detection algorithms are reusing classifiers, such as the deformable parts model (DPM) [4].The DPM first uses classifiers to slide over the image, and then the output of the classifiers is aggregated as the object detection results. This detection method is very time-consuming and tedious, and the detection effect is poor. The current mainstream object detection algorithms mainly use deep learning methods, and we classify them into two categories: two-stage and one-stage. The R-CNN family [5,6,7] is a very classical two-stage algorithm, which first extracts candidate frames, then uses a classifier to filter them, and finally removes duplicate boxes and finetunes the predicted boxes using non-maximal value suppression. The two-stage detector has some advantages in terms of detection accuracy but has disadvantages such as difficulty in training, slow detection speed, and difficulty in optimization. One-stage detectors include methods such as the you only look once (YOLO) series [8,9,10,11,12,13] and single-shot multibox detector (SSD) [14], which use a separate neural network for one forward inference to generate the coordinates and category results of the prediction boxes. The one-stage object detection algorithm improves the speed of detection but loses some of the detection accuracy. In addition, thanks to the excellent performance of the transformer [15] model in the field of natural language processing, Carion et al. introduced the model into the field of computer vision and proposed the detection with transformers (DETR) model [16], which achieved desirable results and provided a new research idea for object detection. However, the objects in natural scenes are multiscale in form, and the objects in the UAV viewpoint are mainly small. So, the mainstream detection algorithms mentioned above are not feasible to be used directly for the object detection task in UAV aerial photography scenes.

A large amount of research work has emerged in the field of object detection in UAV aerial photography scenarios in recent years. Luo et al. [17] optimized the detection performance by improving the network module in YOLOv5. The effectiveness of the proposed strategy is verified by numerous datasets, but the detection of small objects is inferior. Zhou et al. [18] solved the problem of monotonous backgrounds in UAV images by using background replacement from the perspective of data enhancement. However, this work was not effective in improving the detection accuracy of small objects. Du et al. [19] designed sparse convolution to optimize the detection head from the perspective of a lightweight model. The sparse convolution reduces the computational effort but decreases the detection accuracy of the model. Deng et al. [20] proposed a lightweight network in order to improve the efficiency of insulator fault detection in transmission lines by UAVs. The method uses YOLOv4 as the baseline model. Firstly, the original backbone network is replaced with MobileNetv3 [21], which effectively reduces the parameters of the model. Secondly, the method also improves the generalization ability of the model by improving the loss function. Finally, the binary particle swarm optimization idea is introduced to reduce the delay of fault detection. Zheng et al. [22] proposed a multispecies oil palm tree detection method called MOPAD, which can detect oil palm trees well and can accurately monitor the growth of oil palm trees. The method combines Faster RCNN, Refined Pyramid Feature (RPF), and Hybrid Balanced Loss Module. MOPAD achieves desirable observations on three large oil palm tree datasets and the average F1 scores outperform other state-of-the-art detection models. Liu et al. [23] proposed a small-target-detection method in the UAV view to reduce the leakage and false detection rate of small targets. The method uses YOLOv3 as the base model, and it optimizes the backbone network by introducing ResNet [24] units and adding convolutional operations to enhance the receptive field of the model. For the current situation of more and more illegal flights of multi-rotor UAVs, Liu et al. [25] proposed a novel detection method to improve the detection accuracy of multi-rotor UAVs. The method uses Efficientlite to replace the backbone network of YOLOv5, which reduces the complexity of the model and improves the detection efficiency. In addition, adaptive spatial feature fusion is used at the head of the baseline model to optimize the detection accuracy of the model. To improve the detection performance of UAV aerial images, Wang et al. [26] proposed a lightweight detection model called MFP-YOLO by optimizing YOLOv5. Firstly, the method designs a multiplexed inverted residual block (MIRB) and introduces the convolutional block attention module (CBAM) [27], which effectively improves the model’s detection effect under an environment of scale variation and background complexity. Secondly, the formula introduces a parallel convolutional spatial pyramid pooling framework, which takes into account targets at different scales. Finally, a lightweight decoupled detection header is applied to the baseline model, which reduces the parameters of the model while maintaining its detection accuracy. Liu et al. [28] proposed a multi-branch parallel feature pyramid network (MPFPN) to reduce the leakage rate of small-target detection in UAV images. Meanwhile, supervised spatial attention module (SSAM) was added to this network to suppress the interference of background noise. Finally, the effectiveness of the method was demonstrated at the public data level. Most of the current research methods generally have low accuracy for object detection in UAV aerial photography scenarios, and it is difficult to balance the relationship between the accuracy of the model and resource consumption.

To alleviate the above problems, we propose an object detection model based on UAV aerial photography scenarios, called UAV-YOLOv8, using YOLOv8 as the backbone network. This model not only improves the performance of target detection but also does so without too much resource consumption. The main contributions of this paper are as follows:We propose an efficient and fast feature processing module called the FFNB based on the FasterNet block [29]. Utilizing this module, we design two new detection scales that enable the comprehensive fusion of shallow and deep features, significantly reducing the missed detection rate of small objects.We introduce a low-computational-cost dynamic sparse attention mechanism BiFormer [30] in the backbone network, which improves the model’s attention to the critical information in the feature map and optimizes the detection performance of the model.We incorporate WIoU v3 [31] in our bounding box regression loss, which employs a dynamic non-monotonic mechanism to design a more reasonable gradient gain allocation strategy. WIoU v3 effectively reduces the gradient gain of high-quality samples and low-quality samples, which enhances the model’s localization performance and generalization ability.Compared with some mainstream YOLO series models as well as six other classical detection models, the experimental results demonstrate the superiority of our method. In addition, we perform visual analysis from three perspectives to explain that our proposed method effectively improves the detection performance for small objects.

## 2. YOLOv8 Detection Algorithm

The YOLO model has been a great success in the field of computer vision; based on this, researchers have improved and added new modules to the method, proposing many classical models. YOLOv8 is an algorithm released by the Ultralytics company on 10 January 2023. Compared to previous excellent models in the YOLO series (such as YOLOv5 and YOLOv7), YOLOv8 is an advanced and cutting-edge model that offers higher detection accuracy and speed.

The YOLOv8 network structure mainly consists of a backbone, neck, and head, as shown in Figure 1.

### 2.1. Backbone

YOLOv8 uses modified CSPDarknet53 [10] as the backbone network, and the input features are down-sampled five times to obtain five different scale features, in turn, which we denote as B1–B5. The structure of the backbone network is shown in Figure 1a. The Cross Stage Partial (CSP) module in the original backbone network is replaced by the C2f module, and the structure of the C2f module is shown in Figure 1f (n denotes the number of bottlenecks). The C2f module adopts a gradient shunt connection to enrich the information flow of the feature extraction network while maintaining a light weight. The CBS module performs a convolution operation on the input information, followed by batch normalization, and finally activates the information stream using SiLU to obtain the output result, as shown in Figure 1g. The backbone network finally uses the spatial pyramid pooling fast (SPPF) module to pool the input feature maps to a fixed-size map for adaptive size output. Compared with the structure of spatial pyramid pooling (SPP) [32], SPPF reduces the computational effort and has lower latency by sequentially connecting the three maximum pooling layers, as shown in Figure 1d.

### 2.2. Neck

Inspired by PANet [33], YOLOv8 is designed with a PAN-FPN structure at the neck, as shown in Figure 1b. Compared with the neck structure of YOLOv5 and YOLOv7 models, YOLOv8 removes the convolution operation after up-sampling in the PAN structure, which maintains the original performance while achieving a lightweight model. We use P4-P5 and N4-N5 to denote the two different scales of features in the PAN structure and FPN structure of the YOLOv8 model, respectively. Conventional FPN uses a top-down approach to convey deep semantic information. The FPN enhances the semantic information of the features by fusing B4-P4 and B3-P3, but some object localization information will be lost. To alleviate this problem, PAN-FPN adds PAN to FPN. PAN enhances the learning of location information by fusing P4-N4 and P5-N5 to realize path enhancement in a top-down form. PAN-FPN constructs a top-down and bottom-up network structure, which realizes the complementarity of shallow positional information and deep semantic information through feature fusion, resulting in feature diversity and completeness.

### 2.3. Head

The detection part of YOLOv8 uses a decoupled head structure, as shown in Figure 1e. The decoupled head structure uses two separate branches for object classification and predicted bounding box regression, and different loss functions are used for these two types of tasks. For the classification task, binary cross-entropy loss (BCE Loss) is used. For the predicted box bounding regression task, distribution focal loss (DFL) [34] and CIoU [35] are employed. This detection structure can improve detection accuracy and accelerate model convergence. YOLOv8 is an anchor-free detection model that concisely specifies positive and negative samples. It also uses the Task-Aligned Assigner [36] to dynamically assign samples, which improves the detection accuracy and robustness of the model.

## 3. Method

YOLOv8 is a state-of-the-art object detection model and takes into account the multiscale nature of objects, using three scale-detection layers to accommodate objects of different scales. However, the images acquired by UAVs have the problems of complex backgrounds and a high proportion of small objects. This results in the detection structure of YOLOv8 not meeting the detection requirements in UAV aerial photography scenarios. To mitigate the above problems, this paper uses YOLOv8 as the base model and optimizes the model from the perspectives of loss function, attention mechanism, and multiscale feature fusion. The main ideas of the improvement strategy are as follows:

First, WIoU v3 is utilized as the bounding box regression loss. WIoU v3 incorporates a dynamic non-monotonic mechanism and designs a sensible gradient gain allocation strategy, which reduces the occurrence of large or harmful gradients from extreme samples. WIoU v3 focuses more on samples of ordinary quality, thereby improving the model’s generalization ability and overall performance.

Then, the dynamic sparse attention mechanism BiFormer is introduced into the backbone network. BiFormer reduces computation and memory consumption by filtering out most of the low-relevance regions in the feature graph and then applying attention to the high-relevance features. BiFormer improves the model’s attention to the key information in the input features and optimizes the detection performance of the model.

Finally, the efficient feature processing module FFNB is proposed based on FasterNet. FFNB has fewer computation and memory accesses during feature processing. Based on FFNB, we design two new detection layers. Our proposed multiscale feature fusion network makes the shallow feature and deep feature fully complement each other, which effectively improves the detection effect of the model on small objects.

We refer to the final improved network model as UAV-YOLOv8, and the overall framework of the model is shown in Figure 2. To show the structure of the improved model more concisely and intuitively, we omit the drawing of the C2f module and SPPF module in Figure 2. In Figure 2, large, medium, small, X-small, and XX-small are used to represent the size of the object. The improved model changes from the original 3-scale detection to 5-scale detection, which effectively improves the overall detection performance of the model, especially for small objects.

For ease of reading, we give a notation of the terms involved with the formulas in this paper, as shown in Table 1.

### 3.1. Improved Loss Function

The object detection task in the UAV aerial photography scene has a high proportion of small objects, so a reasonably designed loss function can significantly improve the detection performance of the model. YOLOv8 uses DFL and CIoU to calculate the regression loss of the bounding box, but CIoU has the following drawbacks: first, CIoU does not consider the balance of difficult and easy samples. Second, CIoU uses the aspect ratio as one of the penalty factors of the loss function, and if the aspect ratio of the real box and the predicted box are the same, but the values of width and height are different, the penalty term cannot reflect the real difference between these two boxes. Third, the calculation of the CIoU formula involves an inverse trigonometric function, which will increase the consumption of model arithmetic power. The formula of CIoU is shown in Equation (1):(1)$$L_{CIoU}=1-IoU+\frac{\rho^{2}(b,b^{gt})}{{\left(c_{w}\right)}^{2}+{\left(c_{h}\right)}^{2}}+\frac{4}{\pi^{2}}(\tan^{-1}\frac{w^{gt}}{h^{gt}}-\tan^{-1}\frac{w}{h})$$

In Equation (1), Intersection over Union (IoU) denotes the intersection ratio of the prediction box and the real box. Some of the parameters involved in Equation (1) are shown in Figure 3. $\rho (b,b^{gt})$ denotes the Euclidean distance between the centroids of the real box and the prediction box; h and w denote the height and the prediction box; $h^{gt}$ and $w^{gt}$ denote the height and the width of the real box; $c_{h}$ and $c_{w}$ denote the height and the width of the minimum enclosing box formed by the prediction box and the real box.

EIoU [37] improves based on CIoU by treating the length and width separately as penalty terms, reflecting the difference in width and height between the real box and the predicted box, which is more reasonable compared with the penalty term of CIoU. The formula of EIoU is shown in Equation (2):(2)$$L_{EIoU}=1-IoU+\frac{\rho^{2}(b,b^{gt})}{{\left(c_{w}\right)}^{2}+{\left(c_{h}\right)}^{2}}+\frac{\rho^{2}(w,w^{gt})}{{\left(c_{w}\right)}^{2}}+\frac{\rho^{2}(h,h^{gt})}{{\left(c_{h}\right)}^{2}}$$

Some of the parameters involved in Equation (2) are shown in Figure 3; $\rho (w,w^{gt})$ and $\rho (h,h^{gt})$ denote the Euclidean distance of width and Euclidean distance of height between the real box and the prediction box, respectively; $\left(c_{b_{x}},c_{b_{y}}\right)$ and $\left(c_{b_{x}}^{gt},c_{b_{y}}^{gt}\right)$ denote the coordinates of the center points of the real box and the prediction box, respectively.

SIoU [38] introduces the angle between the predicted box and the real box as a penalty factor for the first time. Firstly, based on the magnitude of the angle (as in Figure 3, θ and α) between the predicted box and the real box, the predicted box rapidly moves towards the nearest axis and then regresses towards the real box. SIoU reduces the degrees of freedom of the regression and speeds up the convergence of the model.

While several mainstream loss functions introduced above take a static focusing mechanism, WIoU not only considers the aspect, centroid distance, and overlap area but also introduces a dynamic non-monotonic focusing mechanism. WIoU applies a reasonable gradient gain allocation strategy to evaluate the quality of the anchor box. Tong et al. [31] proposed three versions of WIoU. WIoU v1 was designed with attention-based predicted box loss, and WIoU v2 and WIoU v3 added focusing coefficients.

WIoU v1 introduces distance as a metric of attention. When the object box and the predicted box overlap within a certain range, the penalty of reducing the geometric metric makes the model obtain better generalization ability. The formula for calculating WIoU v1 is shown in Equations (3)–(5):(3)$$L_{WIoUv1}=R_{WIoU}\times L_{IoU}$$
(4)$$R_{WIoU}=\exp(\frac{{\left(b_{c_{x}}^{gt}-b_{c_{x}}\right)}^{2}+{\left(b_{c_{y}}^{gt}-b_{c_{y}}\right)}^{2}}{\left(c_{w}{}^{2}+c_{h}{}^{2}\right)})$$
(5)$$L_{IoU}=1-IoU$$

WIoU v2 is applied to WIoU v1 by constructing the monotonic focus coefficient $L^{*}{}_{IoU}$, which effectively reduces the weight of simple examples in the loss value. However, considering that $L^{*}{}_{IoU}$ decreases as $L_{IoU}$ decreases during model training, resulting in slower convergence, the average of $L_{IoU}$ is introduced to normalize $L^{*}{}_{IoU}$. The formula of WIoU v2 is shown in Equation (6):(6)$$L_{WIoUv2}={\left(\frac{L^{*}{}_{IoU}}{\bar{L_{IoU}}}\right)}^{\gamma }\times L_{WIoUv1},\gamma >0$$

WIoU v3 defines the outlier β to measure the quality of the anchor box, constructs a non-monotonic focus factor r based on β, and applies r to WIoU v1. A small value of β indicates a high anchor box quality, and a smaller r is assigned to it, reducing the weight of high-quality anchor frames in the larger loss function. A large value of β indicates a low-quality anchor box, and a small gradient gain is assigned to it, which reduces the harmful gradients generated by low-quality anchor boxes. WIoU v3 uses a reasonable gradient gain allocation strategy to dynamically optimize the weight of high- and low-quality anchor boxes in the loss, which makes the model focus on the average quality samples and improves the overall performance of the model. The WIoU v3 formulas are shown in Equations (7)–(9). δ and α in Equation (8) are hyperparameters that can be adjusted to fit different models.
(7)$$L_{WIoUv3}=r\times L_{WIoUv1}$$
(8)$$r=\frac{\beta }{\delta \alpha^{\beta -\delta }}$$
(9)$$\beta =\frac{L^{*}{}_{IoU}}{\bar{L_{IoU}}}\in [0,+\infty )$$

By comparing the several mainstream loss functions above, we finally introduce WIoU v3 in the object bounding box regression loss. On the one hand, WIoU v3 takes into account some advantages of EIoU and SIoU, which is in line with the design concept of the excellent loss function. On the other hand, WIoU v3 uses a dynamic non-monotonic mechanism to evaluate the quality of anchor boxes, which makes the model focus more on anchor boxes of ordinary quality and improves the model’s ability to localize objects. For the object detection task in the UAV aerial photography scene, the high proportion of small objects increases the detection difficulty, and WIoU v3 can dynamically optimize the loss weights of small objects to improve the detection performance of the model.

### 3.2. Efficient Attention Mechanism

Due to the complex backgrounds and high proportion of small objects in images captured by UAVs, many detection models have a poor ability to suppress background information. To make the detection model focus more on the key information in the input features and less on the background information, we introduce a dynamic sparse attention mechanism called BiFormer in the backbone network of the model. BiFormer utilizes query adaptation to first filter out the most irrelevant key–value pairs in coarse-grained regions of the input feature graph, efficiently find key–value pairs with high relevance, and then perform attention computation. This significantly reduces computation and storage consumption and enhances the model’s perception of the input content.

YOLOv8 is a convolutional neural network (CNN) model, and a CNN is essentially local processing, so the relationships between global features cannot be obtained. Compared with traditional CNN models, the transformer uses an attention mechanism to obtain the degree of correlation between data and other data, with the property of a global sensing field. An effective attention mechanism can build robust and powerful data-driven models, making the models more flexible when dealing with complex and large data. The attentional mechanism works as follows: first, $\left[x_{1},x_{2},x_{3},\cdots ,x_{T}\right]$ is acquired by encoding the input data sequence $\left[a_{1},a_{2},a_{3},\cdots ,a_{T}\right]$. Then, three matrices of queries Q, keys K, and values V are obtained by linear transformation matrices $W^{Q}$, $W^{K}$, and $W^{V}$, respectively. The dot product between the query and the corresponding key is computed, then normalized, and finally multiplied with matrix V to obtain the weighted sum. $\sqrt{d_{K}}$ is introduced to prevent the gradient of the result from vanishing, and $d_{K}$ denotes the dimensionality of the matrix K. The formula for attention is shown in Equation (10):(10)$$Attention(Q,K,V)=softmax(\frac{QK^{T}}{\sqrt{d_{K}}})V$$

However, the conventional attention mechanism has the drawbacks of high computational complexity and large memory usage. Detection models deployed on UAV platforms are resource-constrained, and if a conventional attention module is introduced directly into the model, it will occupy most of the platform resources and reduce the inference speed of the model. To ease the computational and memory problems, researchers have proposed to reduce resource consumption by replacing global queries with sparse queries that focus on only some key–value pairs. Since then, many related works based on this research idea have appeared, such as local attention, deformable attention, and expansive attention, but all of them are manually produced static patterns and content-independent sparsity. To solve these problems, Lei Zhu et al. [30] proposed a novel dynamic sparse attention: the Bi-Level Routing Attention, whose workflow is shown in Figure 4a.

From Figure 4a, it can be seen that the input feature map $X\in R^{H\times W\times C}$ is firstly divided into S×S subregions, and each region contains $\frac{HW}{S^{2}}$ feature vectors. We change the shape of X to obtain $X^{r}\in R^{S^{2}\times \frac{HW}{S^{2}}\times C}$. Then, the feature vectors are linearly transformed to derive the three matrices, Q, K, and V. The calculation formulae are shown in Equations (11)–(13):(11)$$Q=X^{r}W^{Q}$$
(12)$$K=X^{r}W^{k}$$
(13)$$V=X^{r}W^{V}$$

Then, the attention relation from region to region is obtained by constructing a directed graph to locate its related regions for a given region. The specific implementation process is as follows: Q and V for each region are processed by region averaging to obtain the region level $Q^{r}$ and $K^{r}\in R^{S^{2}\times C}$. Then, the dot product of $Q^{r}$ and $K^{r}$ is calculated to obtain the adjacency matrix $A^{r}\in R^{S^{2}\times S^{2}}$, which is used to measure the inter-region correlation, and the formula is shown in Equation (14):(14)$$A^{r}=Q^{r}{\left(K^{r}\right)}^{T}$$

Next, $A^{r}$ is pruned. The least relevant token in $A^{r}$ is filtered out at the coarse-grained level, and the top k most relevant regions in $A^{r}$ are retained to obtain the routing index matrix, $I^{r}\in N^{S^{2}\times k}$. The calculation formula is shown in Equation (15):(15)$$I^{r}=topkIndex(A^{r})$$

Subsequently, token-to-token attention is used at the fine-grained level. For queries in the region i, this attention is focused only on the k routing regions where $I_{\left(i,1\right)}^{r},I_{\left(i,2\right)}^{r},\ldots ,I_{\left(i,k\right)}^{r}$ are indexed and collects all the K and V tensors in these k regions to acquire $K^{g}$ and $V^{g}$. The calculated formula are shown in Equations (16)–(17):(16)$$K^{g}=gather(K,I^{r})$$
(17)$$V^{g}=gather(V,I^{r})$$

Finally, the collected $K^{g}$ and $V^{g}$ are processed with attention, and a local context enhancement term LCE(V) is added to obtain the output tensor O. The formula is shown in Equation (18):(18)$$O=Attention(Q,K^{g},V^{g})+LCE(V)$$

The BiFormer block is designed based on Bi-Level Routing Attention, as shown in Figure 4b. The DWConv in this block denotes deep separable convolution, which can reduce the number of parameters and computation of the model. LN denotes layer normalization processing, which can accelerate the training and improve the generalization ability of the model. MLP denotes a multilayer perceptron, which further processes and adjusts the attention weights to enhance the model’s attention to different features. The add symbol in Figure 4b indicates connecting two feature vectors.

In this paper, the BiFormer block is introduced into the backbone network. On the one hand, BiFormer can take into account the limited computing power and storage resources of the UAV hardware platform. On the other hand, the dynamic attention mechanism in this block can improve the model’s attention to the vital information of the object and optimize the detection performance of the model. To make full use of the efficient attention mechanism in this block, we use the BiFormer block between B3 and B4 of the model’s backbone network, replacing the original C2f block.

### 3.3. Multiscale Feature Fusion Network

Poor detection of small objects is one of the challenges in object detection tasks in the context of UAV aerial photography. In many existing works [39,40,41,42], detection scales are added to the model to reduce the missed detection rate of small objects, which is an effective improvement method. However, this approach can complicate the structure of the model and increase the consumption of computational and storage resources. To mitigate this problem, a feature processing block called FFNB is proposed in this paper, and a multiscale feature fusion network is designed based on this block. The detection accuracy of small objects is greatly improved while reducing the excessive consumption of resources.

Object detection tasks for UAV platforms are limited by computational resources, and models with simple structure, low latency, and high data throughput are sought. Some classical lightweight networks, such as MobileNet [43], ShuffleNet [44], and GhostNet [45], are using deep convolution or group convolution to extract the spatial features of images. Deep convolution reduces the input of feature dimensions by convolving the input images grouped in feature dimensions, reducing the number of parameters while keeping the feature information largely unchanged. Group convolution can be seen as a sparse form of traditional convolution, where the input channels are convolved one by one, which can be used to reduce the model parameters and achieve the purpose of lightweight models. Most of these lightweight models focus on reducing the number of floating-point operations (FLOPs), and very little related work considers the low floating-point operations per second (FLOPS) of the model. However, the reduction in model parameters does not translate exactly into an increase in the computational speed of the model. Therefore, some work using deep convolution or group convolution in an attempt to design lightweight and fast neural network blocks, in some cases, does not speed up the model operation and even exacerbates the latency.

For an input feature of size h×w×c, the required FLOPs using regular convolution of size k×k are shown in Equation (19). The c in Equation (19) represents the number of channels of the input data.
(19)$$FLOPs_{Conv}=h\times w\times k^{2}\times c^{2}$$

The deep convolution kernel performs a sliding operation on the input channel space to derive the output channel features, and the FLOPs for deep convolution are calculated as shown in Equation (20):(20)$$FLOPs_{DWConv}=h\times w\times k^{2}\times c$$

The depth convolution computation process is shown in Figure 5a.

The popular deep convolution effectively reduces the parameters of the model. But in practical application, depth convolution needs to be followed by additional point-by-point convolution or other computational costs to compensate for the reduction in accuracy after the convolution operation. This introduces additional memory access costs and increases latency. To relieve the above problem, Chen et al. [29] proposed a simple and efficient convolution: partial convolution (PConv). PConv uses regular convolution to perform a convolution operation on some of the continuous features in the input channel, and the remaining features are processed by identity mapping, keeping the channel unchanged. We perform the convolution calculation for the first consecutive feature with channel number $c_{p}$ in the input features, as shown in Figure 5b, and derive the formula for calculating the FLOPs of PConv as shown in Equation (21):(21)$$FLOPs_{PConv}=h\times w\times k^{2}\times c_{p}{}^{2}$$

If $c_{p}$ is 1/4 of the number of input feature channels c, the FLOPs of PConv are only 1/16 of the conventional convolution. This convolution reduces the number of memory accesses while reducing the parameters, and efficiently extracts the spatial features of the input information. Chen et al. proposed the FasterNet block based on PConv, which is a module consisting of a PConv layer and two 1 × 1 convolutional layers connected sequentially, as shown in Figure 6a. The add symbol in Figure 6a indicates connecting two feature vectors. The overuse of normalization and activation layers may lead to a reduction in feature diversity, which may affect the performance of the model. Therefore, the FasterNet block uses normalization and activation layers only after the second convolutional layer. FasterNet block has a simple structure and a low number of parameters for faster operation.

In this paper, we revisit the structure of the FasterNet block. A 1 × 1 convolutional layer is used in this block, which can reduce the number of parameters, speed up the training and increase the nonlinear fitting ability of the model. However, the receptive field of 1 × 1 convolution is relatively small and lacks in acquiring global features. It is also considered that the FasterNet block uses only one shortcut connection, and the input features are convolved through three layers, in turn, which may lead to network degradation and feature disappearance as the depth of the model continues to deepen. To solve the above problems, the FFNB is proposed in this paper based on the FasterNet block, and the structure is shown in Figure 6b.

First, PConv is used to replace the two 1 × 1 convolutional layers in the FasterNet block, which improves the receptive field while making the original module faster and more efficient. Second, the residual concatenation is added to the last two convolutional layers in the block to enrich the features of the output information, reduce the loss of effective features, and optimize the detection performance of the model.

Most of the current mainstream object detection models use convolutional neural networks to extract object features. As the number of convolutions increases, the semantic information of the input features gradually becomes richer, but the detailed features will become fewer and fewer, which is one of the main reasons for the low detection accuracy of many object detection models for small objects. Although YOLOv8 uses a multiscale detection method, it still cannot meet the detection needs of UAV aerial photography scenarios, which leads to the unsatisfactory detection accuracy of the model for small objects.

To improve the detection accuracy of small objects and consider the limited resources of the platform, this paper uses the efficient FFNB to design the feature fusion network. We add two new detection scales to the original three detection scales of YOLOv8 and fuse the shallow information of B1 and B2, which are richer in location information. The FFNB is added as a feature processing block between B1 and B2 of the backbone network, and then the original C2f block between B2 and B3 is replaced using this block. The introduced FFNB can ease the resource consumption caused by multiscale feature fusion. The improved model achieves a five-scale detection, as shown in Figure 7, which effectively improves the detection performance of the model.

## 4. Experiments

### 4.1. Experiment Introduction

This section first introduces the dataset used in this paper, then introduces the experimental environment and training strategy, and finally introduces the evaluation metrics related to the experimental results.

#### 4.1.1. Dataset

The VisDrone2019 dataset [46] is one of the mainstream UAV aerial photography datasets, which was collected and developed by Tianjin University and the data mining team AISKYEYE. The dataset is framed in more than a dozen different cities in China and uses a variety of UAVs for multi-angle, multiscene, and multi-task photography, making the dataset very informative, including the category of detection objects (monotonous and rich), the number of detection objects (fewer and more numerous), the distribution of detection objects (sparse and dense), and the light intensity (day and night). Some representative images in the dataset are shown in Figure 8.

VisDrone2019 contains 10 different types of objects, such as pedestrians, cars, bicycles, etc. Figure 9 shows the information related to the manual labeling of the objects in this dataset. We present the subfigures in Figure 9 in order from left to right and top to bottom. The first subfigure shows the number of objects of each type in the dataset, and indicates that the objects are dominated by cars and pedestrians. The second subfigure shows the size of the object bounding boxes in the dataset, and the coordinates of the centers of all object boxes are fixed at one point. The size of the object bounding box size shows that the dataset contains a large number of small-area objects. The third subfigure is the distribution of the coordinates of the center points of the object bounding boxes, and it can be seen that the center points of the objects are mainly concentrated in the middle and right below the area of the image data. The fourth subfigure is a scatter plot of the corresponding width and height of the object bounding box, with the darkest color at the bottom left of the plot. It further shows that the current dataset is dominated by small objects.

From the above introduction and analysis of the VisDrone2019 dataset, it can be surmised that the dataset contains a large number of small objects, and they exist mainly in a dense and uneven distribution. Compared with the dataset of traditional computer vision tasks, this dataset is a large UAV dataset with multiple scales, scenes, and angles, which is more challenging than the general computer vision tasks.

#### 4.1.2. Experimental Environment and Training Strategies

The hardware platform and environmental parameters used in the experimental training phase are shown in Table 2.

To facilitate flexible deployment on hardware devices in various application scenarios, the YOLOv8 model has been adapted to generate five different scaled models by adjusting two parameters: width and depth. These models are referred to as YOLOv8n, YOLOv8s, YOLOv8m, YOLOv8l, and YOLOv8x. The parameters and resource consumption of the five models increase sequentially, and the detection performance becomes better and better. The width, depth, and maximum number of channels corresponding to these five models are shown in Table 3.

To better study and improve the models, we choose YOLOv8s as the baseline model. Some of the key parameter settings during model training are shown in Table 4.

Under the dataset division rules of the VisDrone 2019 Challenge, this paper divides the dataset into training sets (6471 images), testing sets (1610 images), and validation sets (548 images). To accelerate the model convergence, Mosaic data enhancement is turned off in the last 10 epochs of the training process.

#### 4.1.3. Evaluation Indicators

To test the detection performance of our proposed improved model, we use precision, recall, mAP0.5, mAP0.5:0.95, number of model parameters, model size, and detection speed as evaluation metrics. The following parameters are used in the formulae for some of the above evaluation metrics: TP (predicted as a positive sample and actually as a positive sample as well), FP (predicted as a positive sample, though it is actually a negative sample), and FN (predicted as a negative sample, though it is actually a positive sample). Intersection over Union (IoU) represents the ratio of intersection and concatenation between the bounding box and the true box.

Precision is the ratio of the number of positive samples predicted by the model to the number of all detected samples and is calculated as shown in Equation (22):(22)$$\mathrm{Precision}=\frac{TP}{TP+FP}$$

Recall is the ratio of the number of positive samples correctly predicted by the model to the number of positive samples that actually appeared. Recall is calculated as shown in Equation (23):(23)$$\mathrm{Recall}=\frac{TP}{TP+FN}$$

The average precision (AP) is equal to the area under the precision–recall curve and is calculated as shown in Equation (24):(24)$$\mathrm{AP}=\int_{0}^{1}\mathrm{Precision}(\mathrm{Recall})d(\mathrm{Recall})$$

Mean average precision (mAP) is the result obtained by the weighted average of AP values of all sample categories, which is used to measure the detection performance of the model in all categories, and the formula is shown in Equation (25):(25)$$\mathrm{mAP}=\frac{1}{N}\sum_{i=1}^{N}AP_{i}$$

The $AP_{i}$ in Equation (25) denotes the AP value with category index value i, and N denotes the number of categories of the samples in the training dataset (in this paper, N is 10). mAP0.5 denotes the average accuracy when the IoU of the detection model is set to 0.5, and mAP0.5:0.95 denotes the average accuracy when the IoU of the detection model is set from 0.5 to 0.95 (with values taken at intervals of 0.5).

### 4.2. Experiment Results

#### 4.2.1. Comparative Experiment of Loss Function

To verify the superiority of introducing WIoU v3, we conducted comparison experiments on YOLOv8s using WIoU v3 and some mainstream loss functions, keeping the other training conditions consistent. The experimental results are shown in Table 5. The experimental results show that the model achieves the best detection performance when using WIoU v3 as the bounding box regression loss. In addition, the model’s mAP50 when using WIoU v3 is 0.7% higher than when using CIoU, demonstrating the effectiveness of introducing WIoU v3.

#### 4.2.2. Comparison with YOLOv8

To demonstrate the detection performance improvement effect of the improved model, we conducted comparison experiments between the improved model and the baseline model YOLOv8s. Table 6 shows the AP values for each category and the mAP0.5 values for all categories for the improved model and YOLOv8s. As shown in the comparison results in Table 6, the mAP value of the improved model is improved by 7.7%. The AP values of all the categories are improved to different degrees, and the AP values of three categories (pedestrian, people, and motor) are improved by more than 10%. This indicates that the improved model can effectively increase the detection accuracy of small objects and improve the detection performance.

Figure 10 shows the change curves of some important evaluation metrics of our proposed model and YOLOv8s during the training process. From Figure 10, we can see that our proposed model outperforms YOLOv8s in three detection metrics: precision, recall, and mAP0.5 after about 15 epochs of training from the beginning, and our model starts to stabilize after about 50 epochs of training. Compared with YOLOv8s, our method is faster to train and better to detect.

To further demonstrate the effectiveness of the proposed method, we compared the proposed model with some different sizes of YOLOv8 (YOLOv8n, YOLOv8s, YOLOv8m, and YOLOv8l) on the VisDrone2019 dataset. The experimental results are shown in Table 7. According to the data in Table 7, compared with other models, the improved model has the highest values of the three evaluation indexes Recall, mAP0.5, and mAP0.5:0.95, and the detection performance is better than the model with a larger size than itself. From the experimental results, it can be seen that our proposed five-scale detection structure can improve the detection accuracy of small objects. In addition, the low-computing-power attention mechanism BiFormer, introduced by us, improves the detection performance of the model without consuming too many resources.

#### 4.2.3. Adding BiFormer Block

To obtain the best performance and facilitate the subsequent ablation experiments after adding the BiFormer block to the model, the following comparison experiments were conducted in this study. We used YOLOv8s after the introduction of WIoU v3 as the baseline model and replaced the C2f module with the BiFormer block in different layers of the backbone network, and attained the experimental results shown in Table 8. The + in Table 8 indicates that the BiFormer block is added to the baseline model. B3-BiFormer indicates that the C2f module between layer B3 and layer B4 is replaced using the BiFormer block, and B4-BiFormer indicates that the C2f module between layer B4 and layer B5 is replaced using the BiFormer block. According to the experimental results in Table 8, the best detection performance is achieved when the C2f module between layers B4 and B5 of the baseline model is replaced by the BiFormer module. Compared with the baseline model, mAP0.5 is improved by 0.5%. When adding the BiFormer block, the model can better focus on the essential information in the input features.

#### 4.2.4. Ablation Experiments

To verify the effectiveness of each improvement strategy proposed in this paper, we performed ablation experiments on the baseline model using the VisDrone2019 dataset, and the experimental results are shown in Table 9. B2-FFNB and B1-FFNB in Table 9 indicate the use of FFNB module-based detection layers that fuse the shallow features of B2 and B1 layers, respectively. √ indicates that this improved strategy was used.

The experimental results in Table 9 show that each improvement strategy improved the detection performance to different degrees when applied to the baseline model. WIoU v3 is introduced in the prediction box regression loss. WIoU v3 improves the localization ability of the model using a smarter sample allocation strategy, resulting in improved mAP50 by 0.7%. BiFormer was introduced into the backbone network, replacing the C2f module in the original model. The efficient attention mechanism in BiFormer improves the attention to the key information in the feature map, which increases the mAP50 by 0.5%. The relatively simple structure of the BiFormer module reduces the size of the model by 0.4 MB and the parameters by about 0.2 M (million). The B2-FFNB detection layer was added to the baseline model, increasing the mAP by 4.3%. The currently used dataset contains a large number of small targets, so adding a detection scale to fuse the shallow (B2) feature information can effectively reduce the missed detection rate of small objects. Our proposed efficient and fast FFNB module alleviates the computational and memory burden of feature fusion and reduces the parameters of the model. The B1-FFNB detection layer was added to the baseline model, resulting in the growth of mAP by 2.2%. The model fuses the B1 layer features with richer detail information, which makes the shallow and deep information fully fused and further reduces the missed detection rate of small objects. Our proposed feature fusion network effectively improves the detection performance of the model.

The average detection accuracy of the improved model is increased by 7.7%, most of the detection metrics are effectively improved, and the size and number of parameters of the model are lower than those of the baseline model. However, the addition of two detection layers causes the model structure to become complex and the inference time to become longer.

#### 4.2.5. Comparison Experiments

To show the superiority and effectiveness of the improved algorithm proposed in this paper, we conducted two sets of comparison experiments. In the first set of comparison experiments, we compared the proposed model with some YOLO series algorithms. In the second set of comparison experiments, we compared the proposed model with other excellent models.

The YOLO family of algorithms used in this paper are as follows: YOLOv3 [10], which uses multiscale detection for the first time, and its lightweight version, YOLOv3-tiny; YOLOv4 [11] uses the idea of CSPNet [48] to construct a new backbone network structure called CSPDarknet53, which is lightweight while maintaining the detection accuracy; YOLOv5 uses Mosaic data augmentation to increase the model training speed and detection accuracy. In addition, YOLOv5 also uses the Focus structure, which reduces the computation and parameter count of the model while maintaining the detection performance. YOLOv7 uses the efficient network architecture ELAN [13]. The results of the comparison experiments are shown in Table 10.

According to the experimental results in Table 10, the earlier YOLO series algorithms, such as YOLOv3 and YOLOv4, which have complex structures and large parameter counts, are not conducive to deployment on unmanned platforms and have relatively low detection accuracy. YOLOv3-Tiny achieves a lightweight model but loses a large portion of the detection accuracy. YOLOv5s and YOLOv8s have smaller model sizes, fewer parameters, and better detection performance than previous YOLO versions. However, both models use three-scale detection structures, which cannot fulfill the detection needs of a high proportion of small objects, and therefore result in lower model detection accuracy than the proposed method in this paper at the same size. Compared with UAV-YOLOv8s, YOLOv7 has an advantage in detection speed, but it is deficient in model size and detection performance. In the comparative experimental results, UAV-YOLOv8s has the highest average detection accuracy and the best overall detection performance. The inference time of the improved model increases, but real-time detection can still be achieved.

In this study, a comparative experiment was conducted to evaluate the performance of UAV-YOLOv8s and other mainstream models. The following is a brief introduction to the models used in the comparative experiments: Faster R-CNN [7] optimizes the problems of high computational and structural complexity of R-CNN; Cascade R-CNN [49] proposes a multilevel detection architecture based on R-CNN; RetinaNet [50] proposes Focal loss; CenterNet [51] proposes anchorless frame detection; FSAF [52] algorithm solves the drawbacks of heuristic feature selection; ATSS [53] proposes an adaptive training sample selection mechanism. The experimental results are shown in Table 11.

According to the comparative experimental results in Table 11, the model proposed in this paper has the best detection performance compared to other excellent models. Faster R-CNN, as a two-stage detection algorithm, has a slower detection speed compared to one-stage algorithms Moreover, the resolution of the feature map extracted by the backbone network is small, so it is relatively poor in accuracy for small-object detection. RetinaNet and ATSS focus on how to determine the positive and negative samples. CenterNet removes the computational effort due to the use of anchors, but there are situations such as dense scenes and occlusion of small objects that can lead to some items being missed if multiple prediction centroids are overlapping. The multilevel detection architecture proposed with Cascade R-CNN improves the overall detection performance of the model but increases computational complexity and training difficulty. Both RetinaNet and FSAF use multiscale feature fusion to consider objects at different scales. However, in the context of visual tasks where small objects are the primary detection objects, these two model structures cannot address the detection needs. This results in both RetinaNet and FSAF having poorer detection results than the methods proposed in this paper.

Summarizing the results of the above two sets of comparison experiments, the UAV-YOLOv8s proposed in this paper has better detection performance compared to other models. Our proposed multiscale feature fusion network achieves five-scale detection. This structure for small-object detection has some advantages over the models involved in the above comparison experiments, so our detection results are better than the other models. In addition, we introduced WIoU v3 and BiFormer in the baseline model to optimize the model’s localization ability and noise suppression. The improvement strategy we introduced takes into account the consumption of resources and achieves better detection results.

### 4.3. Visualization Analysis

Deep learning models are characterized by poor interpretability, which somehow hinders the development and application of deep learning. To chart the detection effect of the model proposed in this paper intuitively and conveniently, we conducted comparative experiments to analyze the detection performance of the model from three perspectives: confusion matrix, model inference results and heat map. Finally, to verify the generalizability of our method, we conducted inference experiments on self-made data.

To visualize the ability of our method to predict the object categories, we plotted the confusion matrices of UAV-YOLOv8s and YOLOv8s, as shown in Figure 11. The rows and columns of the confusion matrices represent the true and predicted categories, respectively, and the values in the diagonal region indicate the proportion of correctly predicted categories, while the values in the other regions indicate the proportion of incorrectly predicted categories.

As can be seen from Figure 11, the diagonal region of the confusion matrix in UAV-YOLOv8s is darker in color than YOLOv8s, indicating that the ability of our model to correctly predict the object category has been enhanced. However, for small objects such as the bicycle, tricycle, and awning-tricycle, the proportion of objects judged as background is higher, implying that a large portion of these categories are missed during the detection process. The improved model reduces the missed detection rate for these categories, but the percentage of them being correctly predicted is still low. The bicycle category of transportation is small and usually exists in a dense and occluded form, making it difficult to be detected in environments with complex backgrounds.

To visually demonstrate the detection effect of our method, inference experiments using UAV-YOLOv8s, YOLOv8s and YOLOv5s were performed in this study. We chose four representative scenarios, urban roads, public facility places, market intersections, and traffic intersections, as the experimental data. These scenarios contain numerous and diverse small objects, which are suitable for inference experiments. The detection results of the three models are shown in Figure 12.

As shown in the results in Figure 12, compared with YOLOv8s and YOLOv5s, the method proposed in this paper has the optimal detection accuracy for objects at the far end of the field of view. In addition, our method improves the model’s leakage detection rate for occluded and dense objects, which effectively improves the detection performance.

Gradient-weighted class activation mapping (Grade CAM) was utilized to generate heat maps for YOLOv8s and UAV-YOLOv8s [54]. The heat maps visually and easily reflect which areas of the feature map the model is focusing on. The gradient values are obtained by backpropagation of the confidence of the model output categories by Grade CAM, and the pixels with higher gradients in the feature maps are represented by deeper shades of red in the heatmaps. Conversely, the pixels with lower gradients are represented by deeper shades of blue. The experimental results are shown in Figure 13. From Figure 13, we can surmise that YOLOv8s pays poor attention to small objects and is insensitive to distant objects. The model proposed in this paper has better suppression of background noise and pays more attention to small objects. The model’s attention is more focused on the center point of the object, which makes the predicted bounding box more accurate and thus improves the overall detection performance of the model.

To validate the generalizability of our proposed method, representative image data were collected for inference experiments in this study. The image data were mainly collected from scenes on the campus of Wuhan Institute of Technology and various scenes within the Wuhan city area in Hubei Province, China. These scenes contain a large number of small objects, which are challenging for the inference experiments. The results of the inference experiments are shown in Figure 14. Among them, Figure 14a–c show the detection results in the campus scenario, and Figure 14d–f show the detection results in the Wuhan downtown scenario. The results in Figure 14 show that our method exhibits good detection results in a variety of representative scenes such as streets, campuses, and intersections. Some of the detection results in Figure 14 show that our model has almost no missed detection. These experimental results effectively prove the generalizability of our method.

## 5. Conclusions

For the object detection task in UAV aerial photography scenarios, there are problems such as a high proportion of small objects, complex backgrounds, and limited hardware resources. Most existing models suffer from poor detection accuracy and struggle to achieve a balance between detection performance and resource consumption. To optimize the detection performance of the model while considering platform resource consumption, this paper proposes a UAV aerial scene object detection model called UAV-YOLOv8, based on YOLOv8. Firstly, the WIoU v3 loss function is introduced, which incorporates a dynamic sample allocation strategy to effectively reduce the model’s attention to extreme samples and improve overall performance. Secondly, the efficient dynamic sparse attention mechanism BiFormer is integrated into the backbone network, enhancing the model’s focus on critical information in the feature maps and further optimizing detection performance. Finally, the fast and hardware-friendly FFNB is proposed, and two new detection scales are designed based on this block to realize five-scale detection, which fully integrates shallow and deep features and drastically reduces the missed detection rate of small objects. The improved model achieves an average detection accuracy improvement of 7.7% over the baseline model without increasing its size or parameters, leading to significant enhancements in object detection performance. Moreover, the improved model outperforms some classical algorithms of similar types in terms of detection accuracy.

As the improved model adds two detection layers, the structure of the model becomes complex and the feature map size of the shallow layer is relatively large, resulting in different degrees of improvement in the computation and inference time of the model. The FLOPs value of YOLOv8s is 28.7 billion times and that of UAV-YOLOv8s is 53 billion times, which is nearly double the improvement, and there is still room for optimization of computational resource consumption. The detection accuracy of the improved model is still not high for very small objects such as bicycles, and the next major research focus is to continue to optimize the detection accuracy of the model while keeping resource consumption in mind.

## Figures and Tables

**Figure 1 sensors-23-07190-f001:**
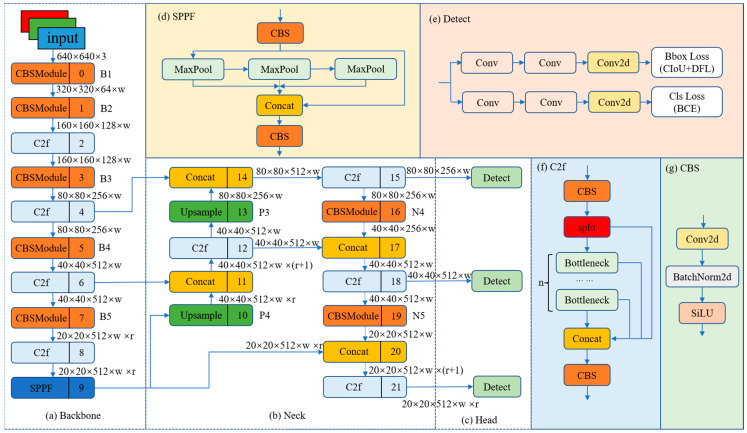
The network structure of YOLOv8. The w (width) and r (ratio) in Figure 1 are parameters used to represent the size of the feature map. The size of the model can be controlled by setting the values of w and r to meet the needs of different application scenarios.

**Figure 2 sensors-23-07190-f002:**
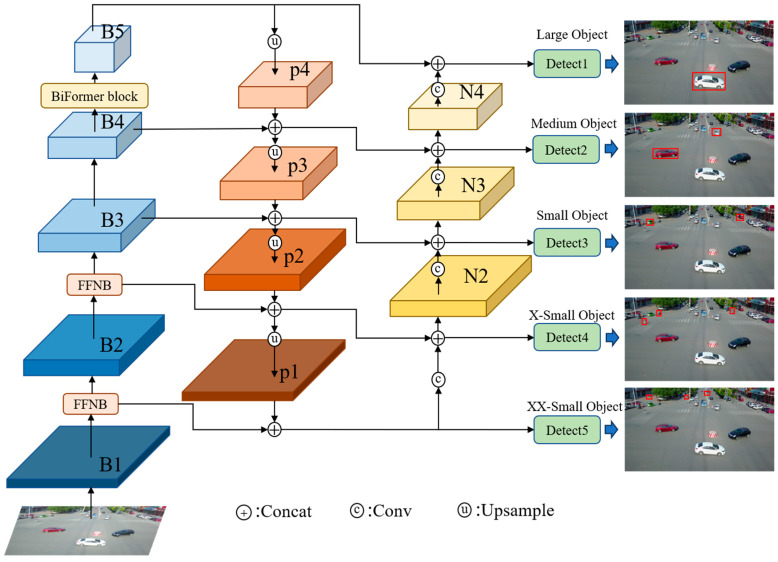
The overall structure of the proposed improved model.

**Figure 3 sensors-23-07190-f003:**
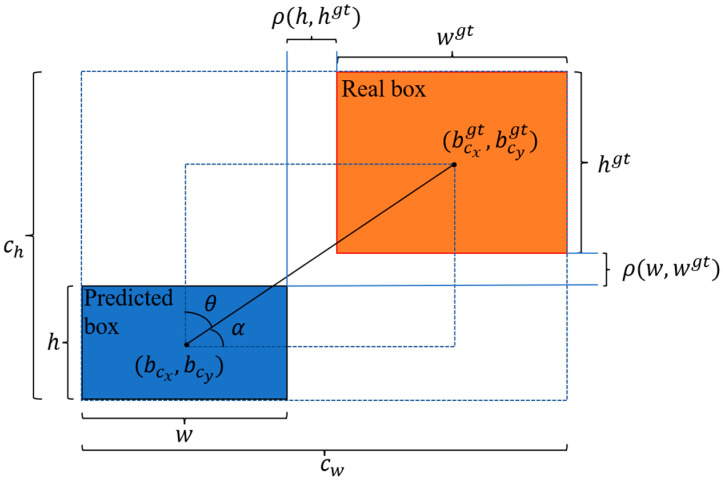
Schematic diagram of the parameters of the loss function.

**Figure 4 sensors-23-07190-f004:**
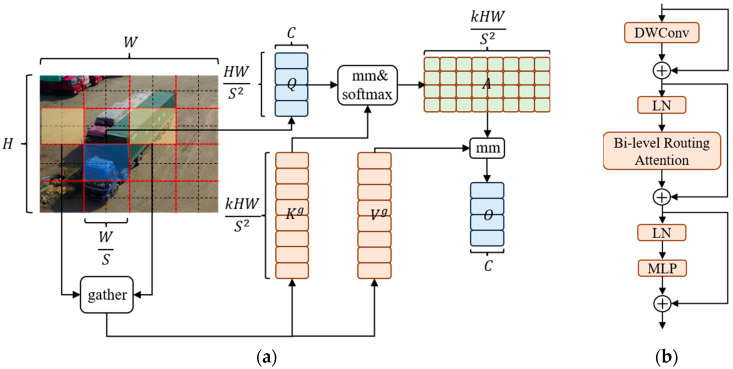
(**a**) Structure of the Bi-Level Routing Attention; (**b**) Structure of the Biformer block.

**Figure 5 sensors-23-07190-f005:**
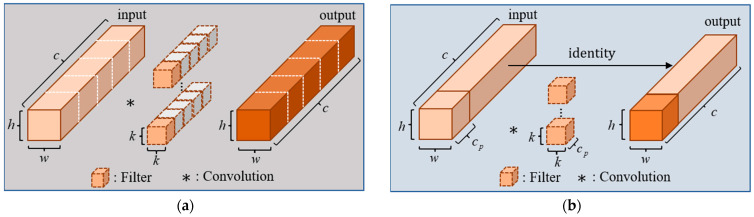
Comparison of DWConv and PConv. (**a**) Structure diagram of deep convolution; (**b**) structure diagram of partial convolution.

**Figure 6 sensors-23-07190-f006:**
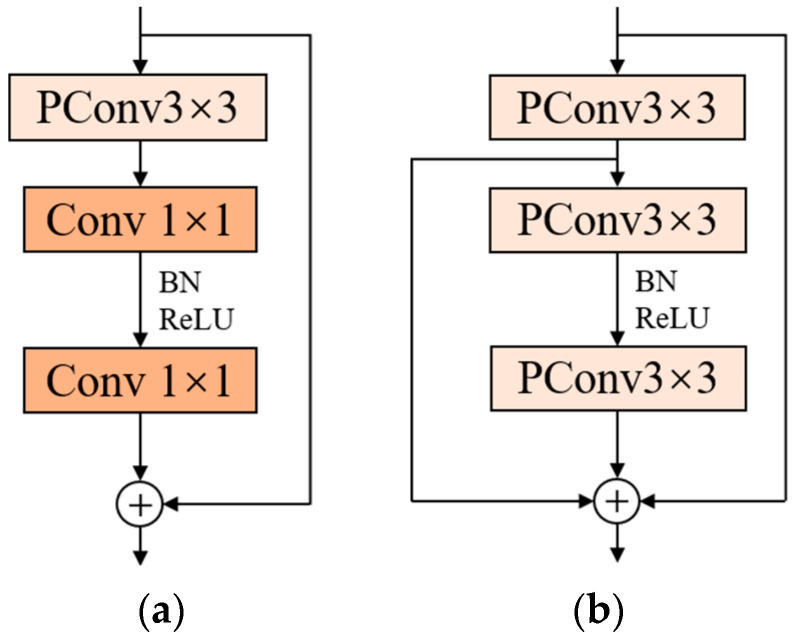
Comparison of FasterNet block and Focal FasterNet block. (**a**) Structure diagram of FasterNet block; (**b**) structure diagram of our proposed module.

**Figure 7 sensors-23-07190-f007:**
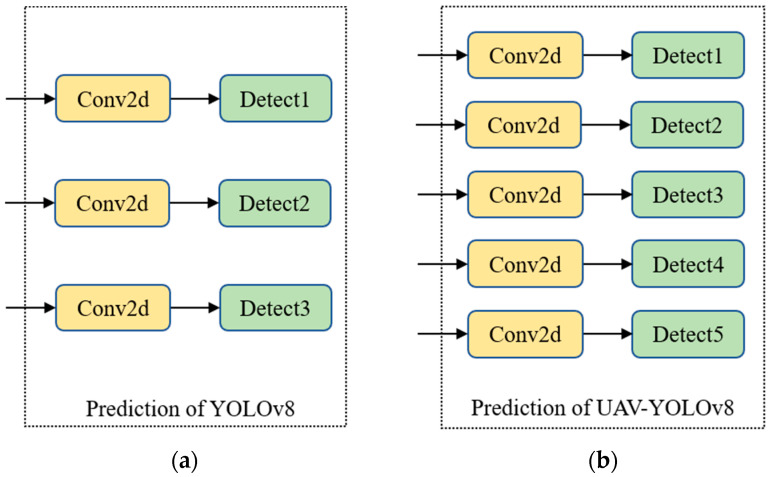
(**a**) Detection head of YOLOv8; (**b**) detection head of our proposed method.

**Figure 8 sensors-23-07190-f008:**
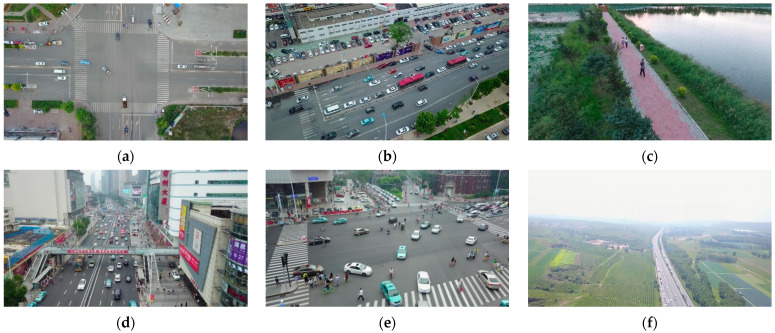
Some representative images from the VisDrone2019 dataset. (**a**) Sparse object distribution; (**b**) Dense object distribution; (**c**) Low number of objects; (**d**) High number of objects; (**e**) Many types of objects; (**f**) Objects are very small; (**g**) Morning; (**h**) Evening; (**i**) Night.

**Figure 9 sensors-23-07190-f009:**
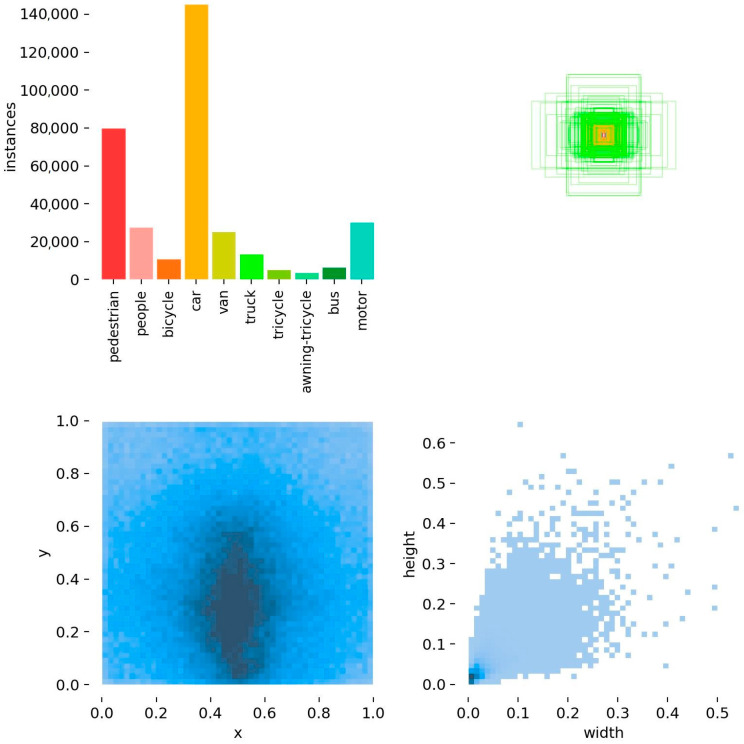
Information about the manually labeling of objects in VisDrone2019 dataset.

**Figure 10 sensors-23-07190-f010:**
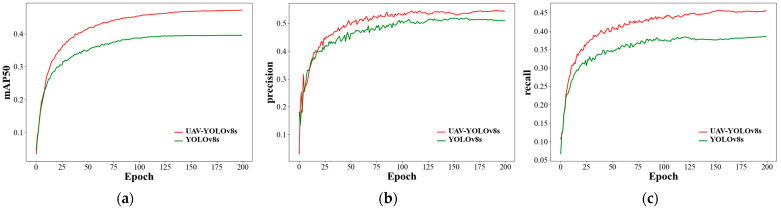
(**a**) Training curve of UAV-YOLOv8s and YOLOv8s in mAP; (**b**) training curve of UAV-YOLOv8s and YOLOv8s in precision; (**c**) training curve of UAV-YOLOv8s and YOLOv8s in recall.

**Figure 11 sensors-23-07190-f011:**
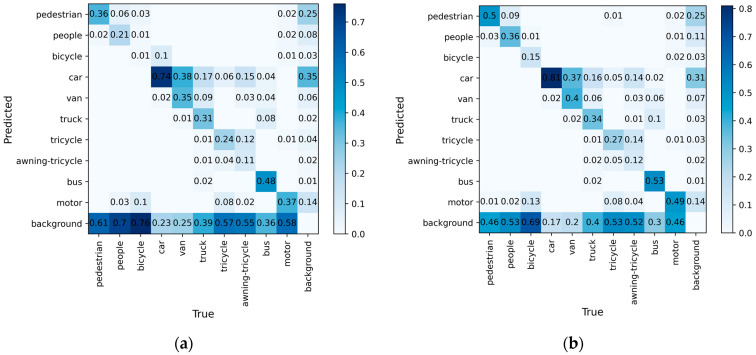
(**a**) Confusion matrix plot of YOLOv8s; (**b**) confusion matrix plot of our model.

**Figure 12 sensors-23-07190-f012:**
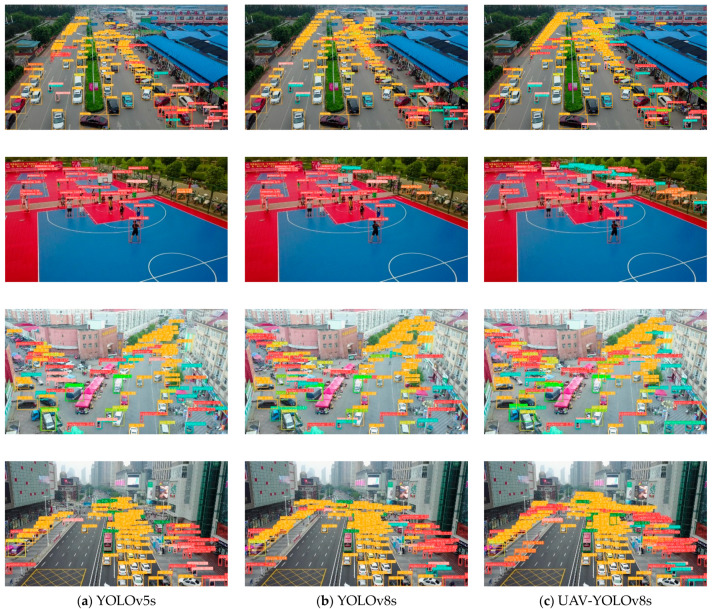
Inference results of three different models on VisDrone2019 dataset. (**a**) Inference results of YOLOv5s; (**b**) inference results of YOLOv8s; (**c**) inference results of our model.

**Figure 13 sensors-23-07190-f013:**
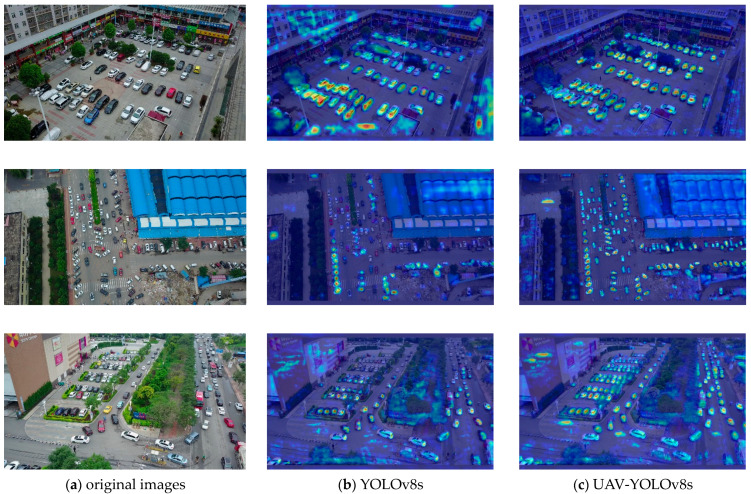
(**a**) Original images; (**b**) heat maps of YOLOv8s; (**c**) heat maps of our model.

**Figure 14 sensors-23-07190-f014:**
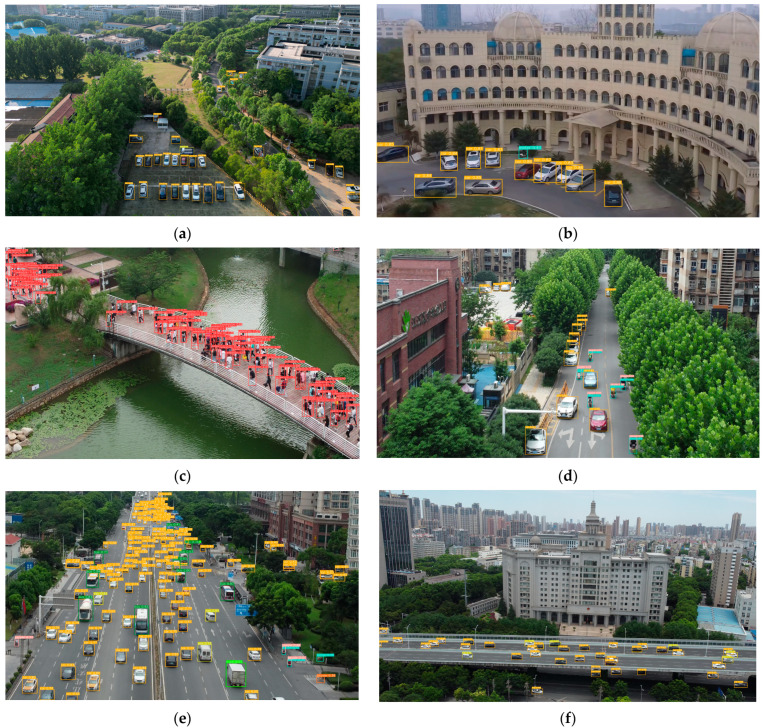
Examples of detection results on self-made data.

**Table 1 sensors-23-07190-t001:** Explanation of notations.

Notation	Explanation
	Loss function
$\left(c_{b_{x}},c_{b_{y}}\right)$, w, h	Denote the coordinates of the center point of the prediction box, width, and height, respectively
$\left(c_{b_{x}}^{gt},c_{b_{y}}^{gt}\right)$, $w^{gt}$, $h^{gt}$	Denote the coordinates of the center point of the real box, width, and height, respectively
$\rho (w,w^{gt})$, $\rho (h,h^{gt})$	Denote the distance of width and height between the real box and the predicted box, respectively
$c_{w}$, $c_{h}$	Denote the width and height of the smallest closed box enclosed by the real and predicted boxes
$L^{*}{}_{IoU}$, r	Monotonic focus coefficient and non-monotonic focus factor
β	Degree of outliers
γ, δ, α	Hyperparameters (we can adjust these hyperparameters to fit different models)
	**BiFormer**
Q, K, V	Queries, keys, and values
$W^{Q}$, $W^{K}$, $W^{V}$	Linear transformation matrices corresponding to Q, K, and V, respectively
$d_{K}$	Dimension of the matrix K
$X\in R^{H\times E\times C}$	Dimension of the input feature map
$A^{r}\in R^{S^{2}\times S^{2}}$	Adjacency matrix
$I^{r}\in N^{S^{2}\times k}$	Routing index matrix
$I_{\left(i,1\right)}^{r},I_{\left(i,2\right)}^{r},\ldots ,I_{\left(i,k\right)}^{r}$	Index values corresponding to the k routing regions with the highest correlation in region i
gather(⋅)	Gathering key–value pairs
Attention(⋅)	Regular attention computation

**Table 2 sensors-23-07190-t002:** Training environment and hardware platform parameters table.

Parameters	Configuration
CPU	i5-12490F
GPU	NVIDIA GeForce RTX 3060
GPU memory size	12G
Operating systems	Win 10
Deep learning architecture	Pytorch1.9.2 + Cuda11.4 + cudnn11.4

**Table 3 sensors-23-07190-t003:** Parameters corresponding to different sizes of YOLOv8.

Model	Depth	Width	Max Channels
YOLOv8n	0.33	0.25	1024
YOLOv8s	0.33	0.50	1024
YOLOv8m	0.67	0.75	768
YOLOv8l	1.00	1.00	512
YOLOv8x	1.00	1.25	512

**Table 4 sensors-23-07190-t004:** Some key parameters set during model training.

Parameters	Setup
Epochs	200
Momentum	0.932
Initial learning rate	0.01
Final learning rate	0.0001
Weight decay	0.0005
Batch size	4
δ (WIoU v3)	1.9
α (WIoU v3)	3
Input image size	640 × 640
Optimizer	SGD
Data enhancement strategy	Mosaic

**Table 5 sensors-23-07190-t005:** Comparison of detection results for different loss functions introduced by YOLOv8s. (The bold data in the table indicate the best results).

Metrics	Precision/%	Recall/%	mAP0.5/%	mAP0.5:0.95/%
CIoU	50.9	38.2	39.3	23.5
DIoU [35]	51.0	38.3	39.5	23.6
GIoU [47]	50.3	38.4	39.6	23.6
EIoU	49.1	38.0	38.7	23.4
SIoU	**51.5**	38.5	39.4	23.4
WIoU v1	50.1	38.5	39.3	23.3
WIoU v2	50.6	38.4	39.3	23.2
WIoU v3	51.3	**38.6**	**40.0**	**23.6**

**Table 6 sensors-23-07190-t006:** Comparison of the proposed improved model and YOLOv8s detection accuracy. (The bold data in the table indicate the best results. All data units in the table are in percent.)

Models	Pedestrian	People	Bicycle	Car	Van	Truck	Tricycle	Awning-Tricycle	Bus	Motor	mAP
YOLOv8s	42.7	32.0	12.4	79.1	44.0	36.5	28.1	15.9	57.0	44.9	39.3
Ours	**56.8**	**44.9**	**18.8**	**85.8**	**50.8**	**39.0**	**33.3**	**19.7**	**64.3**	**56.2**	**47.0**

**Table 7 sensors-23-07190-t007:** Comparative experimental results of the proposed model with four different sizes of YOLOv8. (The bold data in the table indicate the best results.)

Models	Precision/%	Recall/%	mAP0.5/%	mAP0.5:0.95/%	Model Size/MB	Detection Time/ms	Parameter/10^6^
YOLOv8n	43.8	33.0	33.3	19.3	**6.6**	**4.2**	**3.0**
YOLOv8s	50.9	38.2	39.3	23.5	22.5	7.7	11.1
YOLOv8m	56.0	42.5	44.6	27.1	49.6	16.6	25.9
YOLOv8l	**57.5**	44.3	46.5	28.7	83.5	25.6	43.7
Ours	54.4	**45.6**	**47.0**	**29.2**	21.5	19.5	10.3

**Table 8 sensors-23-07190-t008:** Experimental results after introducing the BiFormer block at different layers of the backbone network of the baseline model. (The bold data in the table indicate the best results.)

Model	Precision/%	Recall/%	mAP0.5/%	mAP0.5:0.95/%
Baseline	**50.7**	38.7	40.0	23.6
+B3-BiFormer	49.7	38.6	39.7	23.5
+B4-BiFormer	50.6	**39.2**	**40.5**	**24.1**
+B3-BiFormer+ B4-BiFormer	50.3	39.1	40.2	24.0

**Table 9 sensors-23-07190-t009:** Detection results after the introduction of different improvement strategies. (The bold data in the table indicate the best results.)

Baseline	WIoU v3	BiFormer	B1-FFNB	B2-FFNB	Precision/%	Recall/%	mAP0.5/%	mAP0.5:0.95/%	Model Size/MB	Detection Time/ms	Parameter/10^6^
YOLOv8s					50.9	38.2	39.3	23.5	22.5	7.7	11.1
	√				50.7	38.7	40.0	23.6	22.5	**7.2**	11.1
	√	√			50.6	39.2	40.5	24.1	22.1	7.8	10.9
	√	√	√		**55.8**	42.8	44.8	27.2	21.6	11.2	10.6
	√	√	√	√	54.4	**45.6**	**47.0**	**29.2**	**21.5**	19.5	**10.3**

**Table 10 sensors-23-07190-t010:** Detection results of some YOLO series models and the proposed model. (The bold data in the table indicate the best results).

Models	Precision/%	Recall/%	mAP0.5/%	mAP0.5:0.95/%	Model Size/MB	Detection Time/ms	Parameter/10^6^
YOLOv3	54	43.6	41.9	23.3	213	18.3	103.7
YOLOv3-Tiny	38.2	24.8	23.8	13.3	24.4	2.9	12.1
YOLOv4	36	**48.6**	42.1	25.7	245	25.3	64.4
YOLOv5s	46.4	34.6	34.4	19	**14**	12.0	**7.2**
YOLOv7	51.4	42.1	39.9	21.6	72	**1.7**	64
YOLOv8s	50.9	38.2	39.3	23.5	22.5	7.7	11.1
UAV-YOLOv8s	**54.4**	45.6	**47.0**	**29.2**	21.5	19.5	10.3

**Table 11 sensors-23-07190-t011:** Detection results of the classical model and the proposed model. (The bold data in the table indicate the best results.)

Models	AP0.5/%	AP0.75/%	AP0.5:0.95/%
Faster R-CNN [7]	37.2	22.8	21.9
RetinaNet [50]	19.1	10.5	10.6
Cascade R-CNN [49]	39.1	26.2	24.3
CenterNet [51]	33.7	18.0	18.8
FSAF [52]	36.5	20.6	20.9
ATSS [53]	36.4	23.1	22.3
UAV-YOLOv8s	**47.0**	**30.6**	**29.2**

## Data Availability

Not applicable.

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
