# Peer review of "UAV-YOLOv8: A Small-Object-Detection Model Based on Improved YOLOv8 for UAV Aerial Photography Scenarios"

_sensors, 2023, doi:10.3390/s23167190_

Round 1

Reviewer 1 Report

This research article optimized YOLOv8 and propose an object detection model based on UAV aerial photography scenarios called UAV-YOLOv8, focusing on small object detection. To improve the quality of the paper, the authors need to consider the following items for the revision.

 Major comments:

1.  It is suggested that the author revises English writing.

2. In abstract, reduce the description of the methods and add key findings with drawbacks/scope remain for further study.

3. The introduction need an improvement. There is no detailed literature review for the target objective. Please revise the introduction with review literature based on the objectives.

4.  In method, the description are well explained.

5. As author developed a new approach, it is needed to discuss in depth discussion of the algorithm, how it performs with the current datasets, technical aspects between other algorithms and scope of this algorithm on different datasets in different field and conditions.

Minor comments:

1. The literature [17] optimized the detection (Line 65) can be written Lue et al. [17]… Please check the author guideline and revise similar errors.

2. Figures quality can be improved.

Minor editing of English language required

Author Response

Dear reviewer,

Thank you very much for your time and effort in reviewing our manuscript and for providing valuable suggestions.
We have made point-by-point revisions as per your suggestions.

Please refer to the attached document for details.

Best regards, 
The authors of paper sensors-2547023

Reviewer 2 Report

The submitted article can be recommended for publication after its revision.

For all details see the attached file.

The English style is acceptable. Minor corrections are recommended in the attached file, too.

Author Response

(The authors gave the same response as above.)
